# Adjunctive Effects of a Sub-Antimicrobial Dose of Doxycycline on Clinical Parameters and Potential Biomarkers of Periodontal Tissue Catabolism

**DOI:** 10.3390/dj7010009

**Published:** 2019-01-20

**Authors:** Gülnur Emingil, Ali Gürkan, Taina Tervahartiala, Marcela Hernandez, Semiha Özgül, Timo Sorsa, Saeed Alassiri

**Affiliations:** 1School of Dentistry, Department of Periodontology, Ege University, İzmir 35100, Turkey; aligurkan00@yahoo.com; 2Department of Oral and Maxillofacial Diseases, University of Helsinki, Helsinki University Hospital, 00014 Helsinki, Finland; taina.tervahartiala@helsinki.fi (T.T.); timo.sorsa@helsinki.fi (T.S.); saeed.alassiri@helsinki.fi (S.A.); 3Laboratory of Periodontal Biology and Dentistry, University of Chile, Santiago 8380492, Chile; mhernandezrios@gmail.com; 4Dentistry Unit, Faculty of Health Sciences, Universidad Autόnomia de Chile, Santiago 8910132, Chile; 5School of Medicine, Department of Medical Informatics, Ege University, İzmir 35100, Turkey; semihaozgul@hotmail.com; 6Department of Dental Medicine, Division of Periodontology, Karolinska Institutet, Stockholm 14104, Sweden

**Keywords:** subantimicrobial-dose doxycycline, scaling and root planning, gingival crevicular fluid, adjunctive therapy

## Abstract

**Objectives:** The aim of the present randomized, double-blind, placebo-controlled, parallel-arm study was to examine the effectiveness of a sub-antimicrobial dose of doxycycline (SDD) in combination with nonsurgical periodontal therapy, compared to nonsurgical periodontal therapy alone, on potential gingival crevicular fluid (GCF) biomarkers of periodontal tissue catabolism related to the clinical outcomes over a 12-month period. **Materials and Methods:** GCF was collected and clinical parameters were recorded from 30 periodontitis patients randomized either to an SDD or placebo group. The SDD group received SDD (20 mg) b.i.d for 3 months plus scaling and root planing (SRP), while the placebo group was given placebo capsules b.i.d for 3 months plus SRP. The patients were evaluated every 3 months during the 12-month study period. At each visit, clinical parameters and GCF sampling were repeated. Matrix metalloproteinase (MMP)-8, MMP-9, MMP-13, myeloperoxidase (MPO), osteoprotegerin (OPG), and tartrate-resistant acid phosphatase-5 (TRAP-5) were determined by IFMA and ELISA. **Results:** Significant improvements were observed in all clinical parameters in both groups over 12 months (*p* < 0.0125) while the SDD group showed significantly better reduction in gingival index (GI) and pocket depth and a gain in clinical attachment compared to the placebo group (*p* < 0.05). GCF MMP-8 and OPG levels significantly reduced in the SDD group compared to baseline (*p* < 0.05). GCF MMP-9 significantly decreased in both groups compared to baseline (*p* < 0.05). GCF MPO significantly decreased at 3 and 9 months in the SDD group, while it significantly decreased at 6 months in the placebo group (*p* < 0.05). TRAP and MMP-13 could be detected in none of the samples. **Conclusions:** The present results indicate that three months of adjunctive usage of SDD to nonsurgical periodontal therapy compared to nonsurgical periodontal therapy alone in periodontitis patients results in further improvement of clinical periodontal parameters and GCF markers of periodontal tissue breakdown over a 12-month period. Beneficial effects of adjunctive SDD therapy is likely to be related to the reduced levels of two major periodontitis-associated MMPs, MMP-8 and -9, and their potential oxidative activator MPO.

## 1. Introduction

The upregulated host immune-inflammatory response against the microbial challenge is chiefly mediated by a network of biomarkers that leads to soft- and hard-tissue destruction [1,2]. Matrix metalloproteinases (MMPs)—critical components of the host response to bacteria—are involved in inflammatory and destructive cascades in periodontal pathogenesis [3,4]. Interstitial collagenases, matrix metalloproteinase (MMP)-8, MMP-9, and MMP-13, cleave fibrillar collagens and collagen fragments, subsequently denaturing the collagen into gelatins [5,6,7,8]. Gelatinase MMP-9, found in the tertiary granules of neutrophils, is capable of cleaving gelatins as well as the non-collagenous components of the extracellular matrix and basement membrane [5,7]. These MMPs have also been demonstrated to be secreted by non-leukocytic sources such as epithelial cells, fibroblasts, osteoblasts, and osteoclasts [1,4,6]. MMP-8, MMP-9, and MMP-13 have been demonstrated to be associated with the severity of periodontal disease and are promising candidate biomarkers in oral fluids such as gingival crevicular fluid (GCF), mouthrinse, and saliva [9,10,11,12,13,14,15]. Myeloperoxidase (MPO), a marker of periodontal inflammation, is an oxidative enzyme released from the azurophilic granules of degranulating neutrophils with potent antimicrobial activity [16,17,18]. Besides being an indicator of neutrophil infiltration, the MPO-mediated oxidative pathway can activate proMMPs and contribute to protease activity [19,20,21]. The interaction between collagenases, gelatinases, and MPOs is explained by the oxidative activation of latent collagenases and gelatinases [19,21] and the inactivation of tissue inhibitors of MMPs by the MPO [22]. It has been shown that MPO/MMP-8 could represent the discriminatory biomarkers for the site specific diagnosis of periodontitis, and their association might reflect the disease severity [1,23]. Tartrate-resistant acid phosphatase (TRAP) is a member of the iron-containing purple acid phosphatases and is specifically expressed in osteoclasts and other cells of the monocyte–macrophage lineage [24]. During bone resorption, TRAP is secreted from osteoclasts along with bone matrix products into circulation where its activity reflects the bone resorption rate [25]. Therefore, TRAP is now being investigated as a blood-borne biochemical marker for osteoclastic activity and bone resorption [25,26]. Serum concentrations of TRAP have been demonstrated to be correlated with the rate of bone resorption, while osteoprotegrin administration decreased the serum TRAP-b and reduced osteoclastic bone resorption, which is the hallmark of periodontitis progression [25,27].

Traditional periodontal therapy involves elimination of periodontopathogens by mechanical debridement such as scaling and root planning (SRP) and surgical procedures in conjunction with proper plaque control [28]. Beside the possibility of surgical interventions, non-surgical periodontal therapy may be supported by the use of local or systemic antibiotics. Due to the infective nature of periodontitis, pharmacological agents such as antiseptics and systemic or locally delivered antibiotics have been advocated as adjuncts in the nonsurgical treatment of the periodontal infection [29]. However, because of the increasing resistance to antibiotics, greater efforts are being made to find alternative procedures or new treatment strategies adjunctive to non-surgical periodontal treatment to avoid surgery and the potential drawbacks of antibiotics [30,31,32,33]. 

Pharmacological modulation of the host response represents an opportunity for the management of periodontal disease [34]. Doxycycline’s non-antimicrobial properties is based on the inhibition of the gingival tissues and GCF mammalian collagenase activity of chronic periodontitis [34]. Thereby, both the connective tissue destruction and alveolar bone resorption were shown to be decreased [34]. Given the host modulatory activity of a sub-antimicrobial dose of doxycycline (SDD), numerous studies have evaluated the adjunctive effect of SDD therapy in which SDD was prescribed from 2 weeks to 12 months [35,36,37]. These double-blind, placebo-controlled clinical studies have reported that a combined regimen of SDD and nonsurgical periodontal therapy provides improvements in clinical periodontal parameters in combination with a significant decrease in collagenase activity when compared to nonsurgical periodontal therapy alone [35,36,37]. Long-term usage of SDD did not make the development of TC resistant microorganisms or have any detrimental effect on the periodontal microflora [38,39].

In the present study the adjunctive effects of SDD on both clinical parameters as well as GCF levels of biomarkers of periodontal tissue catabolism, including MPO, MMP-8, MMP-9, MMP-13, and two markers involved in bone turnover, osteoprotegerin (OPG) and TRAP-5, were evaluated in CP patients over a 12-month period and compared with those obtained by nonsurgical periodontal therapy alone. The null hypothesis to invalidate was that after a 1-year follow-up, there were no variations between SRP plus SDD and SRP alone. 

## 2. Material and Methods

### 2.1. Study Population

Thirty chronic periodontitis patients (31–61 years old) were joined from the Department of Periodontology, Dentistry School, Ege University, İzmir, from 2002–2004. All procedures were performed according to the principles outlined in the Declaration of Helsinki and its later amendments or comparable ethical standards on experimentation involving human subjects. Prior to participation the purpose and procedures were fully explained to the patients. They were entered into the study after informed consent was obtained. After complete medical and dental histories were taken, periodontitis was diagnosed according to the clinical and radiographic findings and periodontal history [40]: buccal or oral clinical attachment loss (CAL) ≥3 mm with pocketing >3 mm is detectable at ≥2 teeth [40]. Study patients had at least 14 teeth and had stage III periodontitis. Presence of a history of any systemic disease or a known hypersensitivity to any type of tetracycline were exclusion criteria. Subjects if had received antibiotics or other medicines or periodontal treatment within the past four months were also excluded. Pregnant or lactating women were not included as well. Subjects were also not heavy users of alcohol or tobacco. Smoker patients reported to have smoking less than five cigarettes per day. 

### 2.2. Study Design

This randomized, double-blind, placebo-controlled study of 12-month duration was organized into three phases including screening, treatment, and evaluation. A single examiner (G.E.). evaluated the eligibility and enrollment of the patients into trials. The periodontal status of each patient was assessed at the screening phase. Full mouth probing depth (PD) and CAL were recorded. All measurements were performed with a standard manual probe (Williams probe) at six sites around each tooth. Full mouth gingival index (GI) [41] and plaque index (PI) [42] scores were also recorded. The same examiner unaware of the choice of treatment provided were collected all clinical measurements over the course of the study. Then, subjects were randomized to SDD *(n* = 15) or placebo group (*n* = 15). 

GCF samples were collected at baseline examination. Nonsurgical periodontal therapy was carried out in 4 to 6 sessions by the same clinician. Tooth and root surfaces were instrumented under local anesthesia until they were free of all deposits [43]. All patients were given oral hygiene instruction including tooth brushing and the use of interdental flossing or interdental brushing at each session. The SDD group was given adjunctive SDD (20 mg per capsule), while the placebo group received adjunctive placebo capsules (containing inactive filler; i.e., starch) b.i.d for 3 months. Study medication was advised to be taken once in the morning and once in the evening, 1 h before meals. All patients were represented with a code and were supplied with 2-week doses of SDD or visually identical placebo capsules (28 capsules) in coded bottles. Study medications had indistinguishable in appearance. Obedience was stimulated by giving them in labeled bottles biweekly. 

The evaluation stages included; Recall 1, last day of SDD or placebo therapy (3 months); Recall 2, 3 months after the completion of the medication (6 months); Recall 3, 6 months after the completion of the medication (9 months); Recall 4, 9 months after the completion of the medication (12 months). At each recall visit clinical parameters and GCF sampling were repeated. Regular maintenance therapy was administrated at every visit during the 12-month period and patient was given motivation to reinforce the oral hygiene. Intra-examiner calibration exercise of the examiner gave 97% reproducibility (*κ* = 0.939) within PD and between 2.00 to 10.00 for CAL. 

### 2.3. Gingival Crevicular Fluid (GCF) Sampling

GCF samples were collected from mesiobuccal aspects of a single rooted teeth demonstrating probing depths of 6–8 mm, distributing in different quadrants in each subject., the supragingival plaque was removed from the interproximal surfaces with a curette prior to sampling; these surfaces were dried gently by an air syringe and were isolated by cotton rolls. GCF was collected with filter paper (ProFlow, Inc., Amityville, NY, USA). Paper strips, carefully inserted into the crevice until mild resistance was felt, were left there for 30 s [44]. Attention was taken to prevent mechanical injury. Blood contaminated strips were discarded. The absorbed GCF volume of each strip was determined by an electronic device (ProFlow, Inc., Amityville, NY, USA). Strips were placed into a sterile polypropylene tube and kept at −40 °C until analysis. The readings from the Periotron 8000 were transformed to an actual volume (µL) by reference to the standard curve.

### 2.4. MPO, MMP-9, MMP-13, OPG, and TRAP-5 in GCF by Enzyme-Linked Immunosorbent Assay (ELISA) and MMP-8 by Immunofluorometric Assay (IFMA) Analysis

GCF MMP-9, MMP-13, myeloperoxidase, OPG, and TRAP-5 were determined by ELISA assays according to the manufacturer’s instructions (Immunodiagnostic AG, Bensheim, Germany and Human Biotrac ELISA systems, GE, Healthcare and Amersham, Little Chalfont, UK). GCF samples were assayed at dilutions of 1:10 for MPO and 1:20 for MMP-13. The MMP-13 ELISA detects native, complexed, and fragmented MMP-13 species. The MPO detection limit was 1.6 ng/mL and MMP-13 was 0.032 ng/mL. The level of MMP-8 was assayed at a dilution of 1:4 by IFMA. The detection limit for MMP-8 was 0.08 ng/mL [37]. 

Total MPO, MMP-8, MMP-9, and OPG data were determined by averaging the sampling sites per subjects. Results for MPO, MMP-8, MMP-9, and OPG were converted as total MPO, MMP-9, and OPG (ng/sample) in the GCF sample. 

## 3. Statistical Analysis

Considering a 0.5-mm mean difference between whole mouth PD values of SDD and placebo groups and assuming the standard deviations to be 0.5 in both groups, sample size calculations were performed prior to the study. Accepting a power of 85%, a *p* value of 5%, normal distribution, and equal variances in the SDD and placebo groups, the minimum required sample size per group was calculated as 14 subjects. The primary outcome variable was selected as PD. GCF biomarker levels served as secondary outcome variables. The mean values of the subject whole-mouth clinical parameters and subject mean values for clinical parameters were averaged. The patient was regarded as the unit of analysis. Intragroup comparisons of the clinical parameters between baseline and 3, 6, 9, and 12 months were analyzed by the Friedman test. Bonferroni corrected Wilcoxon signed rank test was used to analyze the significance of the changes over time. The Mann-Whitney test was used to determine significant differences between the SDD and placebo groups. Biochemical variables were tested with the non-parametric model of Brunner-Langer using a web-based software (R software, version 3.3.1, package: nparLD, R Foundation for Statistical Computing, Vienna, Austria; http://r-project.org). 

## 4. Results

### 4.1. Patient Disposition and Demographics

Table 1 presents patient demographics. Gender, ages, and smoking habits of the patients were similar in both groups (*p* > 0.05). 

### 4.2. Clinical Results (Whole Mouth)

The mean values of the subject whole-mouth clinical parameters for the SDD and placebo groups were given in Table A1. Baseline clinical periodontal parameters were similar between both study groups (*p* > 0.05). The periodontal conditions of both SDD and placebo groups markedly enhanced between baseline and the re-examinations at 3, 6, 9, and 12 months (*p* < 0.0125). The whole-mouth mean CAL of the SDD group significantly decreased during the study period (*p* < 0.0125), while the whole-mouth CAL scores of the placebo group showed significant improvement at 3 months (*p* < 0.0125). There were not any significant changes at other time points (*p* > 0.0125). 

SDD group demonstrated a significantly greater reduction in PD than placebo group at the 6, 9, and 12 month visits (*p* = 0.04, *p* = 0.02, and *p* = 0.04, respectively), while both showed a similar reduction in PD at 3 months (*p* > 0.05). The CAL of the SDD group had greater improvement than the placebo group which was significant at 6 and 9 months (*p* = 0.04, *p* = 0.04, respectively). The whole-mouth CAL scores were comparable for both study groups for other time points (*p* > 0.05). GI scores showed statistically significant improvement in SDD compared to the placebo group at 3, 6, and 9 months (*p* = 0.01, *p* = 0.01, and *p* = 0.01, respectively). There was a significant reduction in the PI scores of both groups, but no significant differences detectable between them over the entire study period (*p* > 0.05). 

### 4.3. Clinical Results (Sampling Sites)

The mean of the clinical parameters and GCF values of the study sites were similar between both groups at baseline (*p* > 0.05). Significant improvement was observed at 3 months which was maintained during the 12-month study period compared to baseline (*p* < 0.0125). No significant difference was observed between groups at all time points (*p* > 0.05). (Table A2).

### 4.4. GCF MMP-8, MMP-9, MPO, and OPG Levels

GCF TRAP and MMP-13 levels remained below the detection limit in a considerable part of the study population. MMP-13 could be detected in 15 out of 30 patients and TRAP was detected in 8 of 46 CP patients in the baseline samples.

Figure 1 demonstrates the changes in MMP-8 levels in the GCF samples. SDD and placebo groups had comparable GCF MMP-8 total amount at baseline (*p* > 0.05). GCF MMP-8 total amount significantly decreased in both groups at 12 months compared to baseline (*p* < 0.05). GCF MMP-8 total amount in the SDD group tended to show a larger decrease compared to that of the placebo group at the 6 month mark, but this was not statistically different between groups. GCF MMP-8 total amount was stabilized for the rest of the study period in both groups. 

Figure 2 shows the change in MMP-9 levels in the GCF samples. The SDD and placebo groups had a similar GCF MMP-9 total amount at baseline (*p* > 0.05). GCF MMP-9 total amount significantly decreased in both groups at 12 months compared to baseline (*p* < 0.05). GCF MMP-9 total amount tended to decrease more at the third month in the SDD group compared to placebo group, then a similar decrease pattern was observed in both groups. Then, no additional decrease was observed in either group, although the levels were still significantly lower than the baseline levels. 

Figure 3 outlines the change in MPO levels in the GCF samples. GCF MPO significantly decreased in the SDD and placebo groups at all time points compared to baseline (*p* < 0.05). Similar to the GCF MMP-9 pattern, GCF MPO total amount of the SDD group showed more of a decrease at the third month compared to the placebo group, they were similar at the sixth month, and then the placebo group showed an increased level during the rest of the study. In contrast, GCF MPO levels continued to decrease over 12 months in the SDD group. 

Figure 4 demonstrates the change in OPG levels in the GCF samples. The SDD and placebo groups had a similar GCF OPG total amount at baseline (*p* > 0.05). GCF OPG total amount significantly decreased in the SDD and placebo groups at all time points compared to baseline (*p* < 0.05). The placebo group showed an increased change from the third month to the twelfth month. In contrast, GCF OPG levels continued to decrease over 12 months in the SDD group. 

## 5. Discussion

The efficacy of adjunctive SDD on both clinical parameters and GCF MMP-8, MMP-9, MMP-13, MPO, TRAP, and OPG levels beyond that obtained by nonsurgical periodontal therapy alone in chronic periodontitis patients has been investigated in the present double blind, placebo-controlled, parallel-arm study. Both therapy provided significant improvement in the management of chronic periodontitis, but adjunctive SDD has more clinical benefit than SRP alone.

Both SDD and placebo groups showed significant improvements in clinical parameters, which was evident at 3 months and maintained throughout the study period. All patients received adequate periodontal maintenance therapy at every 3 months after active periodontal therapy [45]. This might improved oral hygiene conditions throughout the study period in both groups. SRP therapy leads to the resolution and control of the inflammatory response, facilitates the elimination of a favorable environment for the colonization of the periodontopathogens, and causes the cessation of the progression of periodontal disease, thereby resulting in a relative gain of clinical attachment and reduction of the probing depth [46,47,48].

In previous studies, SDD was prescribed to chronic periodontitis patients in either combination with scaling and prophylaxis, or SRP therapy [34,35,36,37,38,39,49,50]. Our present findings that adjunctive SDD has more clinical benefit than SRP alone in terms of PD reduction and clinical attachment gain confirm and further extend the findings previously reported [34,35,36,37,38,39,49,50]. In our study, SDD was prescribed for 3 months as an adjunct to SRP. Even though the duration of the drug regimen in our study was shorter than that of previous studies [38,39,50], clinical improvements observed in deep and moderate pockets as early as 3 months after commencing drug therapy were similar to those of previous studies [35,36,38,39,50,51], and importantly, this improvement was maintained until the end of 12-month study period. This clinical improvement in the SDD group might be attributed to the effectiveness of nonsurgical periodontal therapy, as well as due to the long-term beneficial-host modulatory-effect of doxycycline [33,52]. Based on the present data we might suggest that a 3-month treatment duration may be sufficient for producing prolonged and beneficial effects, which might be dependent on the substantivity of SDD [33,53].

Due to the complexity of the periodontitis pathogenesis, the balance between pro-inflammatory and anti-inflammatory mediators is more important than the levels of any single inflammatory mediator in connective tissue and bone catabolism during the disease progression [2]. The levels and especially activation of MMP-8, MMP-9, and MMP-13 have been shown to be correlated with active stages of periodontal disease, and they decrease to levels found in periodontally healthy GCF after conventional periodontal treatment [5,14,15,33]. MPO, by activating MMPs oxidatively in inflamed periodontal tissues, could lead to the upregulation of the MMP-dependent proteolytic cascades and participate in the progression of periodontal disease [10,16,18,23,54,55,56,57]. Studies have shown dramatic elevation of MPO levels in periodontitis, suggesting that MPO could be a useful diagnostic marker for the activity of the acute inflammation [17,58].

In the present study, the SDD and placebo groups had similar GCF MMP-8, MMP-9, and MPO levels at baseline. After adjunctive SDD or SRP therapy, GCF MMP-8, MMP-9, and MPO total amount showed marked reductions throughout the study period. Hence, decreased biomarker levels in both the SDD and placebo groups’ GCF during active periodontal therapy might suggest the effectiveness of nonsurgical periodontal therapy (oral hygiene instruction and SRP) in reducing the bacterial load in the periodontal environment. GCF MMP-8 total amount in the SDD group tended to decrease more compared to the placebo group at the 6 month mark, although the difference was not statistically significant. GCF MMP-8 total amount was stabilized for the rest of the study period in both groups. GCF MMP-9 total amount tended to decrease more at 3 months in the SDD group compared to the placebo group. Then, no additional decrease was observed in the groups, although the levels were still significantly lower than the baseline levels. Similar to MMP-9, GCF MPO total amount of the SDD group showed more of a decrease at 3 months compared to the placebo group, but they were similar at 6 months. Thereafter, those levels increased for the rest of the study periods in the placebo group while the GCF MPO levels continued to decrease over the 12 months in the SDD group. Based on the present data we might suggest that while mechanical treatment alone led only to a transient decrease in the levels of these biomarkers, SDD stabilized those enzyme levels for 12 months, although the results were not statistically significant. In a preliminary study, short term (2 week) SDD resulted in the reduction of collagenase activity by approximately 60–80%, which indicated the ability of TCs to decrease the severity of inflammation, thereby blocking the breakdown of the periodontal tissues [34,52]. Our findings are consistent with others and our previous studies which demonstrated that adjunctive SDD therapy can reduce the activity or down-regulate the expression of host collagenases/MMPs by a mechanism unrelated to the antimicrobial efficacy of this drug [37,50,51,52]. Furthermore, Golub et al. [51] showed that SDD was effective in the suppression of pathologically excessive collagenases associated with both inflammatory and bone destructive disease and crucial in the destruction of type I and III collagens found in the periodontal ligament [51]. On the other hand, fibroblast-type collagenase (MMP-1), which is involved in normal connective tissue remodelling, has been shown to be relatively resistant to doxycycline inhibition treatment [59]. This selective MMP inhibition mechanism of doxycycline might provide a safe and effective therapeutic approach for reducing pathologically excessive collagenases, without interfering with the normal connective tissue remodelling required to maintain normal tissue integrity. Lee et al. [49] found that 9 months SDD usage resulted in significantly greater per-patient average in both MMP-8 and MMP-13 activities in the SDD group compared to placebo group. Several studies confirmed that adjunctive SDD treatment following SRP suppresses the activity of the tissue destructive enzymes MMP-8 and MMP-13. At present, quantitative oral fluid chair-side/point-of-care adjunctive diagnostic oral fluid assays are commercially available for MMP-8 [14,15]. In the present study, MMP-13 could be detected in only about 32% of CP patient at baseline, therefore, we were unable to evaluate the effect of SDD therapy in CP patients.

MPO can oxidatively promote collagenolytic catabolism, i.e., active periodontal degeneration with a balance in favor of MMP cascade activation. In the present study, lower GCF MPO levels in the adjunctive SDD group might suggest that SDD in vivo, and in addition to MMPs, can down-regulate the levels of MPO, a unique neutrophil-derived oxidative up-regulator of MMPs, at the periodontitis sites. Our results demonstrate that GCF MMP-8, MMP-9, and MPO levels could be linked to such reduced oxidative and collagenolytic activities due to their impact on MMP-dependent proteolysis of the extracellular matrix.

Several placebo-controlled randomized clinical studies have evaluated the adjunctive effect of SDD therapy on different host response markers in GCF [37,50,60,61,62,63,64,65]. It has been confirmed that SDD therapy inhibits the activity or down-regulates the expression of host inflammatory markers other than MMPs by a mechanism unrelated to the antimicrobial efficacy of this drug [35,37,49]. On the other hand, treatment of periodontitis with sub-antimicrobial doxycycline failed to reduce the GCF osteocalcin levels [51].

In the present study, the adjunctive effect of SDD was evaluated for biomarkers of bone turnover, TRAP and OPG, in GCF for the first time. There are a limited number of studies investigating the TRAP enzyme, whose biological function is not known in periodontal disease. Shibutani et al. [66] by staining TRAP and TRAP^+^ cells observed osteoclastic activity at the alveolar bone surface that is related to the inflammatory changes in the gingiva in an experimentally induced periodontitis model in beagle dogs. Crotti et al. [67] and Da Costa et al. [68] observed increased accumulation of TRAP^+^ cells in CP patients compared to controls and suggested that the inflammatory response triggers osteoclastic differentiation in periodontitis lesions. In another experimental periodontitis study, systemic OPG-Fc sharply decreased serum TRAP-5b levels and stayed low during the study period [27]. GCF TRAP levels were investigated in CP patients, and TRAP could be detected in about 17% of the samples. OPG is one of the molecules that is a negative regulator of bone resorption and limits the duration and extent of the immune and inflammatory responses [69]. Localized absence of OPG in diseased periodontal tissues has been associated with periodontal disease activity and progression [70,71]. In the present study, the GCF OPG total amount significantly decreased in the SDD and placebo groups at all time points compared to baseline. The placebo group showed an increased level from 3 months to 12 months. In contrast, GCF OPG levels continued to decrease over 12 months in the SDD group. To the best our knowledge, this is the first study investigating this bone marker in an SDD group. Previously, Golub et al. [51] observed a significant reduction in pyridoline-containing collagen breakdown fragments in the GCF of CP patients while osteoclacin levels were unchanged after the SDD therapy.

In conclusion, SDD medication in combination with SRP therapy improved clinical periodontal parameters and GCF biomarker levels of chronic periodontitis patients. However, as a limitation of the present study was that GCF TRAP and MMP-13 levels remained below the detection limit in a considerable part of the study population. Taken together, these results demonstrate that the long-term success and efficacy of the SDD treatment is through the MMPs pathway in the periodontal tissues. The present data might further support the effectiveness of the SDD therapy in adjunct to nonsurgical periodontal therapy in the management of chronic periodontitis.

## Figures and Tables

**Figure 1 dentistry-07-00009-f001:**
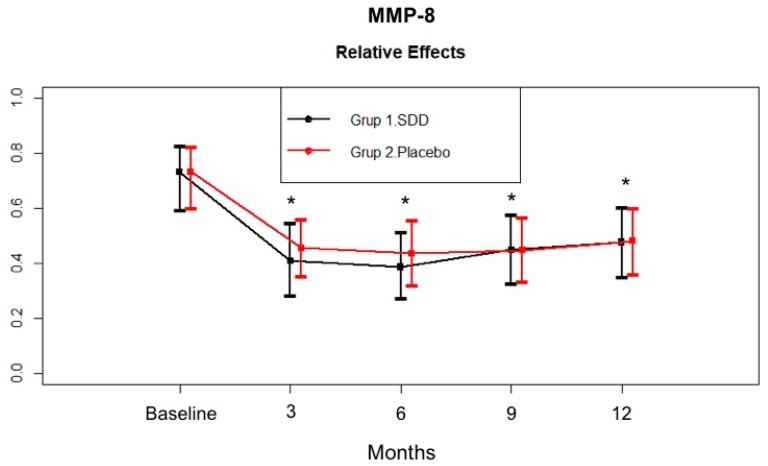
Gingival crevicular fluid (GCF) matrix metalloproteinase (MMP)-8 total amount (pg/site) in the SDD and placebo groups from baseline to 12 months. Relative treatment effects as revealed by Brunner-Langer analysis. * Significant difference from baseline after SDD or placebo therapy (*p* < 0.0125).

**Figure 2 dentistry-07-00009-f002:**
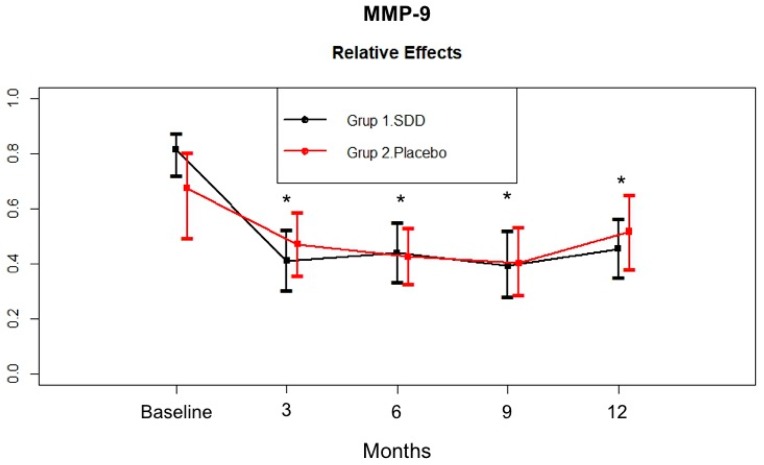
GCF MMP-9 total amount (pg/site) in the SDD and placebo groups from baseline to 12 months. Relative treatment effects as revealed by Brunner-Langer analysis. * Significant difference from baseline after SDD therapy or placebo therapy (*p* < 0.0125).

**Figure 3 dentistry-07-00009-f003:**
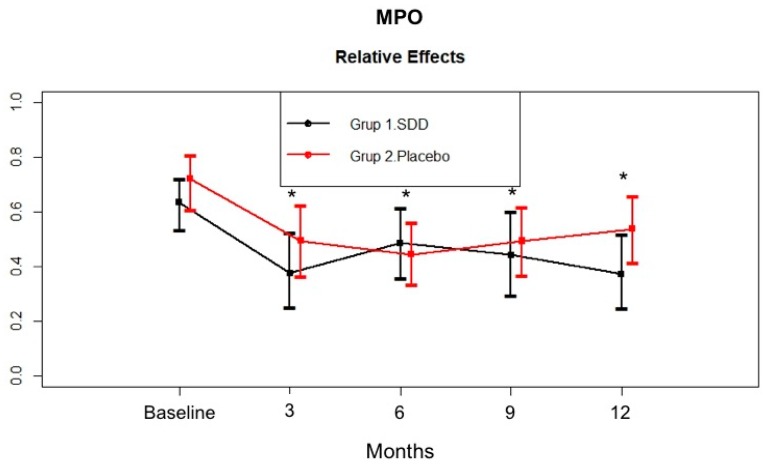
GCF myeloperoxidase (MPO) total amount (pg/site) in the SDD and placebo groups from baseline to 12 months. Relative treatment effects as revealed by Brunner-Langer analysis. * Significant difference from baseline after SDD therapy or placebo therapy (*p* < 0.0125).

**Figure 4 dentistry-07-00009-f004:**
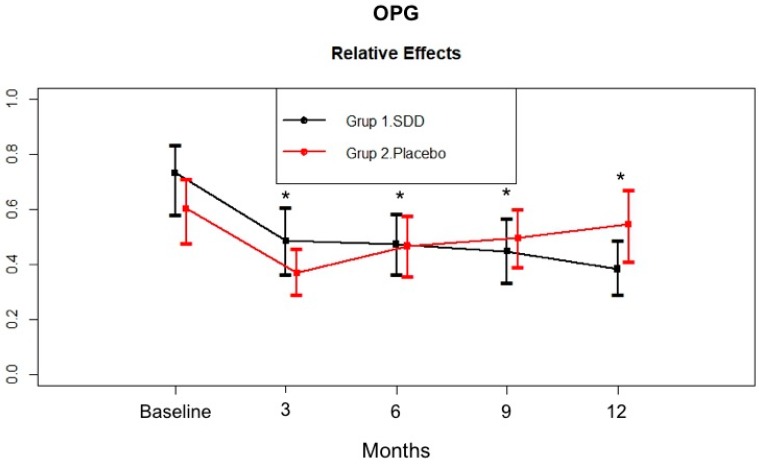
GCF osteoprotegerin (OPG) total amount (pg/site) in the SDD and placebo groups from baseline to 12 months. Relative treatment effects as revealed by Brunner-Langer analysis. * Significant difference from baseline after SDD therapy or placebo therapy (*p* < 0.0125).

**Table 1 dentistry-07-00009-t001:** Demographic characteristics and smoking history of the sub-antimicrobial dose of doxycycline (SDD) and placebo groups.

	SDD Group	Placebo Group
Male/Female	11:4	10:5
Mean age, years	48.9 ± 6.6	49.2 ± 7.3
Age range	38–59	40–61
Smokers (*n*)	7	6

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
