# Peer review of "Adjunctive Effects of a Sub-Antimicrobial Dose of Doxycycline on Clinical Parameters and Potential Biomarkers of Periodontal Tissue Catabolism"

_dentistry, 2019, doi:10.3390/dj7010009_

Round 1

Reviewer 1 Report

In the manuscript entitled: “Adjunctive Effect of Sub-Antimicrobial Dose Doxycycline on Potential Biomarkers of Periodontal Tissue Catabolism” the authors analysed the effectiveness of sub-antimicrobial dose doxycycline (SDD) in combination with nonsurgical periodontal therapy, comparing to nonsurgical periodontal therapy alone, on potential gingival crevicular fluid (GCF) biomarkers of periodontal tissue catabolism related to the clinical outcomes over a 12-month period.

In their study, the authors GCF was collected and clinical parameters were recorded, from 30 chronic periodontitis patients randomized either to SDD or to placebo groups. SDD group received SDD (20 mg) b.i.d for 3 months plus scaling and root planing (SRP), while placebo group was given placebo capsules b.i.d for 3 months plus SRP. The patients were evaluated every 3 months during 12 months study period.

The authors found that significant improvements were observed in all clinical parameters in both groups over 12 months while SDD group showed significantly better reduction in GI, pocket depth, and gain in clinical attachment compared to the placebo group.  TRAP and MMP-13 could be detected in none of the samples.

The authors concluded that adjunctive SDD can improve especially the clinical periodontal outcomes in chronic periodontitis patients. SDD reduced the levels of two major periodontitis associated MMPs, MMP-8 and -9 and their potential oxidative activator MPO.

Moreover, they stated that these biomarkers are thus more reduced by SDD vs placebo indicating that SDD can down-regulate oxidative-proteolytic periodontal tissue destruction cascade associated with beneficial clinical outcomes.

Major comments:

In general, the idea and innovation of this study, regards the analysis of combination therapy sub-antimicrobial dose doxycycline on potential biomarkers of periodontal tissue catabolism is interesting, because the role of these biomaterials in the periodontal repair and regeneration is validated but further studies on this topic could be an innovative issue in this field could be open an innovative matter of debate in literature by adding new information. Moreover, there are few reports in the literature that studied this interesting topic with this kind of study design.

The study was well conducted by the authors; However, there are some concerns to revise that are described below.

The introduction section resumes the existing knowledge regarding the main causes of the aetiology of periodontitis and periodontal defects.

However, as the importance of the topic, the reviewer strongly recommends to update the literature through read, discuss and cites in the references with great attention all of those recent interesting articles, that helps the authors to better introduce and discuss the aim of the study in light of also of the role of the other delivery agents that improve the efficacy of the non-surgical periodontal therapy: 1) Isola G, Matarese G, Williams RC, Siciliano VI, Alibrandi A, Cordasco G, Ramaglia L. The effects of a desiccant agent in the treatment of chronic periodontitis: a randomized, controlled clinical trial. Clin Oral Investig. 2018 Mar;22(2):791-800. doi: 10.1007/s00784-017-2154-7. 2) Tabenski L, Moder D, Cieplik F, Schenke F, Hiller KA, Buchalla W, Schmalz G, Christgau M. Antimicrobial photodynamic therapy vs. local minocycline in addition to non-surgical therapy of deep periodontal pockets: a controlled randomized clinical trial. Clin Oral Investig. 2017 Sep;21(7):2253-2264. 3) Matarese G, Ramaglia L, Cicciù M, Cordasco G, Isola G. The Effects of Diode Laser Therapy as an Adjunct to Scaling and Root Planing in the Treatment of Aggressive Periodontitis: A 1-Year Randomized Controlled Clinical Trial. Photomed Laser Surg. 2017 Dec;35(12):702-709. doi: 10.1089/pho.2017.4288.

The authors should be better specified, at the end of the background section, the rational of the study and the aim of the study with the null hypothesis. Moreover, should better specify, by adding a table, the periodontal results obtained especially for PD, CAL, Gingival recession and Keratinized tissues obtained after therapy in a table.

The discussion section appears well organized with the relevant paper that support the conclusions, even if the authors should better discuss the importance of the periodontal defect in the frontal area with the relative aesthetic problems. The conclusion should reinforce in light of the discussions.

In conclusion, I am sure that the authors are fine clinicians who achieve very nice results with their adopted clinical protocol. However, this study, in my view does not in its current form satisfy a very high scientific requirement for publication in this journal and requests some revisions before a further re-evaluation of the manuscript.

Minor Comments:

Abstract:

-          Better formulate the conclusion section by better specifying the aim of the study

Introduction:

-          Please refer to major comments;

Discussion

-          Please add a specific sentence that clarifies the results obtained in the first part of the discussion

-          last paragraph of the discussion: Please reorganize this paragraph that is not clear

Author Response

Reviewer 1

Major comments:

In general, the idea and innovation of this study, regards the analysis of combination therapy sub-antimicrobial dose doxycycline on potential biomarkers of periodontal tissue catabolism is interesting, because the role of these biomaterials in the periodontal repair and regeneration is validated but further studies on this topic could be an innovative issue in this field could be open an innovative matter of debate in literature by adding new information. Moreover, there are few reports in the literature that studied this interesting topic with this kind of study design.

The study was well conducted by the authors; However, there are some concerns to revise that are described below.

The introduction section resumes the existing knowledge regarding the main causes of the aetiology of periodontitis and periodontal defects.

However, as the importance of the topic, the reviewer strongly recommends to update the literature through read, discuss and cites in the references with great attention all of those recent interesting articles, that helps the authors to better introduce and discuss the aim of the study in light of also of the role of the other delivery agents that improve the efficacy of the non-surgical periodontal therapy: 1) Isola G, Matarese G, Williams RC, Siciliano VI, Alibrandi A, Cordasco G, Ramaglia L. The effects of a desiccant agent in the treatment of chronic periodontitis: a randomized, controlled clinical trial. Clin Oral Investig. 2018 Mar;22(2):791-800. doi: 10.1007/s00784-017-2154-7. 2) Tabenski L, Moder D, Cieplik F, Schenke F, Hiller KA, Buchalla W, Schmalz G, Christgau M. Antimicrobial photodynamic therapy vs. local minocycline in addition to non-surgical therapy of deep periodontal pockets: a controlled randomized clinical trial. Clin Oral Investig. 2017 Sep;21(7):2253-2264. 3) Matarese G, Ramaglia L, Cicciù M, Cordasco G, Isola G. The Effects of Diode Laser Therapy as an Adjunct to Scaling and Root Planing in the Treatment of Aggressive Periodontitis: A 1-Year Randomized Controlled Clinical Trial. Photomed Laser Surg. 2017 Dec;35(12):702-709. doi: 10.1089/pho.2017.4288.

Introduction has been revised and relevant information has been added in view of the reviewer’s suggestion.

The authors should be better specified, at the end of the background section, the rational of the study and the aim of the study with the null hypothesis.

The null hypothesis has been added.

Moreover, should better specify, by adding a table, the periodontal results obtained especially for PD, CAL, Gingival recession and Keratinized tissues obtained after therapy in a table.

In order to better specify the clinical results, table including the clinical data has been placed.

And figures were removed.

The discussion section appears well organized with the relevant paper that support the conclusions, even if the authors should better discuss the importance of the periodontal defect in the frontal area with the relative aesthetic problems. The conclusion should reinforce in light of the discussions.

In conclusion, I am sure that the authors are fine clinicians who achieve very nice results with their adopted clinical protocol. However, this study, in my view does not in its current form satisfy a very high scientific requirement for publication in this journal and requests some revisions before a further re-evaluation of the manuscript.

Minor Comments:

Abstract:

-          Better formulate the conclusion section by better specifying the aim of the study

We have revised as has been suggested.

Introduction:

-          Please refer to major comments;

Discussion

-          Please add a specific sentence that clarifies the results obtained in the first part of the discussion

            A specific sentence has been added in the first part of the discussion

-          last paragraph of the discussion: Please reorganize this paragraph that is not clear

            We tried to do best to revise this paragraph.

Reviewer 2 Report

Thank you for the opportunity to review this novel research study. The idea that adding SDD has a compound effect is powerful. Routine non-surgical cleaning has no impact on the autoimmune nature of periodontal disease and this is an excellent study to highlight the need for addressing periodontal disease with a multi-pronged attack.

On page 3 line 106 you state that enrollment of patients into the trial was determined by a single examiner - but what were the criteria used to determine this?

On page 6 line 204 you talk about a study arm that is 6 months, 9 or 12 months long. Does this mean that some patients were taking doxycycline for 12 months? You should talk about the unintended consequences and risks of this. Research exists that dentists are already one of the few professions where antibiotic prescribing is on the rise - it's falling in all other health professions.

On page 11 line 299 you state that the disease was "similar" in both SDD and placebo. Can you elaborate on this? Perio disease is a condition with strong measurable metrics like periodontal pockets, bleeding scores, attachment loss etc. What criteria were used to categorize as similar?

Overall, I have 3 concerns - two of them minor and one critical:

1. It is very long. I would suggest being more concise - you are currently at 

2. There should be a "Limitations" subheading under which you describe all the limitations of your study. 

3. You have made a statement at the end that ethical approval was gained. This is, by far, the most important concern. It's not clear to me from your statement whether an official institutional approval was gained or if the authors are just saying they feel it met standards. It is customary to state which institution (since your author team represents several institutions) provided this ethical approval to conduct human research and what the ID number of the IRB approval was. Additionally, IRB approval must be mentioned in the Methods section...not just at the end statement.

Author Response

Reviewer 2

Thank you for the opportunity to review this novel research study. The idea that adding SDD has a compound effect is powerful. Routine non-surgical cleaning has no impact on the autoimmune nature of periodontal disease and this is an excellent study to highlight the need for addressing periodontal disease with a multi-pronged attack.

On page 3 line 106 you state that enrollment of patients into the trial was determined by a single examiner - but what were the criteria used to determine this?

            Criteria was the expert of the clinician who is involved in this. We specify the clinician in the text.

On page 6 line 204 you talk about a study arm that is 6 months, 9 or 12 months long. Does this mean that some patients were taking doxycycline for 12 months? You should talk about the unintended consequences and risks of this. Research exists that dentists are already one of the few professions where antibiotic prescribing is on the rise - it's falling in all other health professions.

We actually mean the follow up and visit stages after 3 month SDD usage. In order not to cause any misunderstanding we have revised the sentence.

On page 11 line 299 you state that the disease was "similar" in both SDD and placebo. Can you elaborate on this? Perio disease is a condition with strong measurable metrics like periodontal pockets, bleeding scores, attachment loss etc. What criteria were used to categorize as similar?

            This has been clarified.

Overall, I have 3 concerns - two of them minor and one critical:

1. It is very long. I would suggest being more concise - you are currently at 

            We have tried to do our best to shorten the paper. 

2. There should be a "Limitations" subheading under which you describe all the limitations of your study.

            Limitation subheading has been added.

3. You have made a statement at the end that ethical approval was gained. This is, by far, the most important concern. It's not clear to me from your statement whether an official institutional approval was gained or if the authors are just saying they feel it met standards. It is customary to state which institution (since your author team represents several institutions) provided this ethical approval to conduct human research and what the ID number of the IRB approval was. Additionally, IRB approval must be mentioned in the Methods section...not just at the end statement.

            Although there is no ethics number for this study, all procedures were performed according to the principles outlined in the Declaration of Helsinki and its later amendments or comparable ethical standards on experimentation involving human subjects. The purpose and procedures were fully explained to all patients prior to participation, and patients were entered into the study if only informed consent was obtained from each subject.

This statements have been placed into the methods section.

Reviewer 3 Report

This is a potential interesting manuscript. Some revisions are suggested.

1.Abbreviations should be spelled out when they first appear.

2. While the authors have described the policy of using SDD for therapy, the disadvange and advantage of SDD should be described.

3. section 2.3: How to extract the content from paper strip for ELISA analysis?

4. Legend: Figure 1A?  or Figure 1?

5. SDD (text) or LDD (Figure labeling)?  should be consistent.

6. Whether the clinical efficacy of SDD group is due to oral hygiene (decrease in plaque index in SDD group) or SDD?

7.For MMP-8 ELISA data, total amount of MMP-8 was around 0.2-1 pg/site. This is generally too low for the detection by ELISA kit?   Similarly also the ELISA data of MPO and OPG?   How about the range of detection limit of the ELISA kit?

Author Response

Reviewer 3

This is a potential interesting manuscript. Some revisions are suggested.

1.Abbreviations should be spelled out when they first appear.

            They have been revised as has been suggested.

2. While the authors have described the policy of using SDD for therapy, the disadvange and advantage of SDD should be described.

the advantage of the SDD has been added into the manuscript as has been suggested.

3. section 2.3: How to extract the content from paper strip for ELISA analysis?

Extraction has been done as described by Emingil et al J Periodontol-04, Mäntylä et al J Perio Res-03, 06 and Golub et al  Infl Res-1997 –

4. Legend: Figure 1A?  or Figure 1?

Table and figures have been revised

5. SDD (text) or LDD (Figure labeling)?  should be consistent.

They have been revised.

6. Whether the clinical efficacy of SDD group is due to oral hygiene (decrease in plaque index in SDD group) or SDD?

            Actually for the SDD the clinical efficacy and significant difference from placebo group was observed for whole mouth PD, CAL and GI scores not for PI scores. The sufficient periodontal maintenance therapy including supragingival scaling and oral hygiene instruction given to the study patients at every 3 months after active periodontal therapy which could provide similar PI scores during the whole study period. (This is present in discussion in second paragraph).

7.For MMP-8 ELISA data, total amount of MMP-8 was around 0.2-1 pg/site. This is generally too low for the detection by ELISA kit?   Similarly also the ELISA data of MPO and OPG?   How about the range of detection limit of the ELISA kit?

MMP-8 was not assessed by ELISA but by IFMA as described Sorsa et al Oral dis-10 for examples and Mäntylä et al J Periores-03, 06, MPO et al, see Baeza et al J Clin Perio-16 the details.

Reviewer 4 Report

This is a comprehensive randomized, double-blinded, placebo-controlled clinical trial by Emingil et al.. The study was well-designed and conducted with scientific rigor. The manuscript was also well-written. Some comments as following:

The data also includes clinical parameters, which can also reflect on the title

The introduction was not clear "why would you like to to do this study?" and "how this study will add to the literature and contribute to the field"? (More biomarker targets?)

It will be nice if you can correlate the changes of each biomarker with the improvement of the clinical outcome to help the field consolidate "potential" biomarkers.

Since the new perio classification has come out. Maybe consider to add some new description of the cases in the study population section? 

Figures: a few figures do not have unit over the x-axis; and a bit confusing on using the same "*" for two different t-tests results. (Can also be concised to describe only once in the end)

Does smoker (relatively higher number of subjects in this study) have different outcome and response to the treatment?

Author Response

Reviewer 4

This is a comprehensive randomized, double-blinded, placebo-controlled clinical trial by Emingil et al.. The study was well-designed and conducted with scientific rigor. The manuscript was also well-written. Some comments as following:

The data also includes clinical parameters, which can also reflect on the title

            The clinical parameters have been added to the title.

The introduction was not clear "why would you like to to do this study?" and "how this study will add to the literature and contribute to the field"? (More biomarker targets?)

In the present study the aim was to investigate the effectiveness of SDD on Periodontal Tissue Catabolism markers concomitantly.

It will be nice if you can correlate the changes of each biomarker with the improvement of the clinical outcome to help the field consolidate "potential" biomarkers.

Correlation analysis was performed; no statistical differences have been detected.

Since the new perio classification has come out. Maybe consider to add some new description of the cases in the study population section? 

            We have revised our study population according to the new classification.

Figures: a few figures do not have unit over the x-axis; and a bit confusing on using the same "*" for two different t-tests results. (Can also be concised to describe only once in the end)

Tables were placed instead of figures as has been suggested by the reviewer.

Does smoker (relatively higher number of subjects in this study) have different outcome and response to the treatment?

This is the part of the study published by the same author. Smoking information could not be able to given due to the length of the manuscript. The effects of the smoking was evenly distributed between both groups (data not shown).

Round 2

Reviewer 1 Report

In the R1 version of the manuscript entitled: “Adjunctive Effect of Sub-Antimicrobial Dose Doxycycline on Potential Biomarkers of Periodontal Tissue Catabolism” the authors followed all the issues suggested by the reviewer. Though the changes based on the reviewer comments, almost of the criticisms were carefully analysed and solved. I have carefully evaluated all parts of the manuscript. I believe that the article, in this version, is now adequate for publication in this journal.

Reviewer 3 Report

Accept